# A Gain for Reconstruction, A Pain for Generation: Exploiting Representation in Visual Tokenization

## Abstract

Discrete visual tokenization is a cornerstone of modern auto-regressive (AR) image generation, yet current methods are fundamentally constrained by a trade-off between reconstruction fidelity and semantic expressivity. In this work, we first propose a principled framework for token representation learning based on three pillars: feature alignment with foundation models, structural diversification of the codebook into specialized subspaces, and explicit disentanglement to enforce semantic independence. We materialize these principles in a novel tokenizer, Semantic Subspace Quantization (SSQ), which achieves state-of-the-art image reconstruction. However, this success reveals a critical and previously overlooked paradox: the semantically rich, structured representations that excel at reconstruction cause a significant performance collapse in standard AR generative models. To resolve this Reconstruction-Generation Discrepancy, we introduce a novel tokenizer-generator co-design methodology, systematically adapting the AR model's architecture and training curriculum to harness the multi-faceted nature of SSQ's tokens. Our final, synergistic system effectively alleviates this discrepancy, achieving state-of-the-art performance on high-fidelity reconstruction and generation, demonstrating a new path forward for discrete visual modeling.

## 1 Introduction

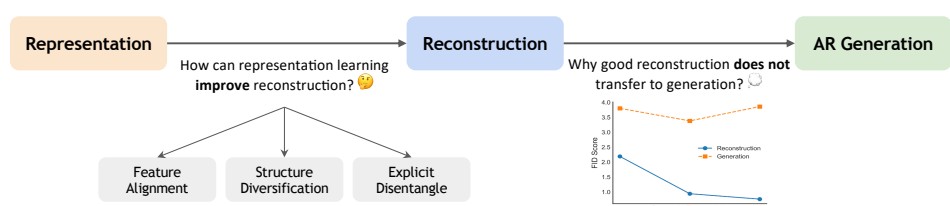

Figure 1: This work aims to systematically understand and exploit the role of representation for visual tokenization and auto-regressive visual generation.

Discrete visual tokenization has become the *de facto* engine for modern image generation (Esser et al., 2021; Lee et al., 2022; Sun et al., 2024), enabling powerful auto-regressive (AR) transformers to model images in the token space, akin to their success in natural language (Brown, 2020; Radford et al., 2021). Architectures like VQGAN (Esser et al., 2021) learn a codebook to quantize image features into a sequence of discrete tokens. The quality of these tokens is paramount, as it directly dictates the upper bound on performance for any downstream generative model.

However, the dominant paradigm of using a monolithic codebook trained on a pixel-reconstruction loss faces a fundamental representation trade-off. The limited size of a single codebook struggles to capture both the fine-grained details necessary for high-fidelity reconstruction and the rich semantic abstractions required for creative generation. Crucially, naively scaling the codebook is not a viable solution; it often leads to diminishing returns, with performance saturating or even degrading for reconstruction and generation (Yu et al., 2024b; Zhu et al., 2024b).

In this work, we first tackle this challenge by asking: *How can we systematically imbue visual tokens with rich semantic representation to improve reconstruction?* We deconstruct this problem and introduce a new conceptual framework built upon three fundamental principles: **1) Feature Alignment**: Directly injecting knowledge from pre-trained foundation models to ensure tokens are semantically grounded. **2) Structural Diversification**: Decomposing the monolithic codebook into specialized, multi-faceted subspaces to capture a wider range of visual concepts. **3) Explicit Disentanglement**: Enforcing semantic independence between these subspaces to create a highly structured and interpretable representational space.

We materialize these principles in a unified tokenizer, Semantic Subspace Quantization (SSQ), which systematically exploits representation to achieve a new state-of-the-art in image reconstruction. This is the gain. However, this victory in reconstruction reveals a deeper, more troubling paradox. When this semantically superior tokenizer is paired with a standard AR generative model, its generation performance unexpectedly gets worse. This is the pain.

As strikingly visualized in Fig. 1, as SSQ's reconstruction FID decreases (lower is better), the generation FID sharply increases. We identify this paradox as the **Reconstruction-Generation Discrepancy**—a fundamental tension largely hidden because prior works primarily evaluated on latent diffusion models (Yao et al., 2025) or specialized Visual Auto-regressive (VAR) models (Li et al., 2024b; Qu et al., 2024). The very structural diversity that makes SSQ excel at reconstruction complicates the implicit assumptions of sequential modeling in generic AR architectures.

This discovery compels a paradigm shift from a decoupled to an integrated approach. To resolve this paradox, we introduce a novel tokenizer-generator co-design strategy. We design the AR model's architecture and training procedure to explicitly accommodate the structured, multi-subspace nature of our advanced tokenizer, effectively teaching the generator to leverage the rich representation it is given. Our contributions are multi-faceted:

- We establish a new principled framework for representation learning in visual tokenization, based on the core pillars of alignment, diversification, and disentanglement.

- We propose SSQ, a novel tokenizer that embodies this framework and sets a new state-of-the-art in image reconstruction.

- We discover and analyze the Reconstruction-Generation Discrepancy, a fundamental paradox where improvements in token quality lead to a degradation in AR generation.

- We deliver a novel co-design strategy that effectively alleviates this gap, creating a synergistic tokenizer-generator system that achieves sota performance on both reconstruction and generation.

## 2 RELATED WORK

**Visual Tokenizers.** Toward building expressive visual tokenizers, one intuitive strategy is enlarging codebook size. However, research (Sun et al., 2024; Zhu et al., 2024b; Shi et al., 2024) indicates that naively expanding the codebook size can lead to performance saturation or even degradation. Subsequent efforts then resort to enhancing the representation of latent tokens. We empirically categorize them following the three principles as elaborated above. First, on feature alignment, VQGAN-LC (Zhu et al., 2024b) uses pre-trained feature clusters to implicitly regularize the large codebook, helping to maintain a higher usage rate. VA-VAE (Yao et al., 2025) directly aligns the visual token embeddings with features from a vision foundation model. Similar idea has been explored from different perspectives (Gu et al., 2024; Zhu et al., 2024a; Yu et al., 2024a; Qu et al., 2024; Lin et al., 2025; Xiong et al., 2025; Li et al., 2025; Wu et al., 2024; Liang et al., 2024; Wang et al., 2024b). Second, on structural diversification, representative works include ImageFolder (Li et al., 2024b), TokenFlow (Qu et al., 2024), UniTok (Ma et al., 2025), etc. However, these methods were developed without a systematic framework aimed at maximizing token representation, but with different primary motivations that distinguish them from our work. E.g., ImageFolder utilizes two codebooks to improve computational efficiency, while TokenFlow and UniTok focus on creating a unified token for multimodal understanding and generation. Our work, SSQ, is the first to formalize these disparate efforts into a unified and principled framework designed explicitly for representation learning. Furthermore, to the best of our knowledge, we are the first to introduce explicit disentanglement as a third critical principle, proposing a regularization term that encourages different subspaces to capture distinct semantic factors.

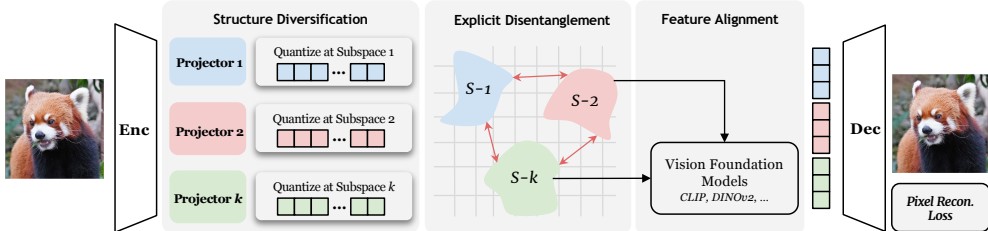

Figure 2: SSQ framework for exploiting representation learning in visual tokenizers, including three pillars: Structure Diversification, Explicit Disentanglement, and Feature Alignment.

**Auto-regressive (AR) Visual Generation** uses a next-token prediction approach to sequentially generate images or videos. VQGAN (Esser et al., 2021), a pioneering model, utilizes a transformer to predict tokens sequentially. RQ-VAE (Lee et al., 2022) extends VQGAN by incorporating a residual tokenization mechanism and adding an AR transformer head to predict residual tokens at a finer depth. Inspired by the success of AR generation in large language models, recent efforts seek to scale this paradigm in visual generation. LlamaGen (Sun et al., 2024) extends the VQGAN transformer architecture to the Llama (Touvron et al., 2023) framework, demonstrating promising scaling behaviors. VAR (Tian et al., 2024) designs a specialized next-scale prediction paradigm for visual generation, reducing auto-regressive steps and enhancing performance. Even though, standard GPT-style AR model is still a popular choice in recent works (Yu et al., 2024c; Pang et al., 2025; Hu et al., 2025), as its scaling potential has been largely proved in language domain.

## 3 SEMANTIC SUBSPACE QUANTIZATION

**Preliminary** VQGAN (Esser et al., 2021) employs a learnable discrete codebook $\mathbf{C} \in \mathbb{R}^{K \times D}$ to represent images, where $K$ is the codebook size while $D$ is the dimensionality of the codes. Given an input image $x$, the encoder transforms it into a latent feature $h = \text{Enc}(x)$. Then, the closest codebook entry for each patch is retrieved from the codebook to serve as the quantized representation:

$$q^i = \text{Quant}(h^i, \mathbf{C}) := \underset{c^j \in \mathbf{C}}{\arg \min} \left\| h^i - c^j \right\|, \tag{1}$$

where $h^i \in \mathbb{R}^D$, $c^j \in \mathbb{R}^D$, $q^i \in \mathbb{R}^D$ denotes the encoded latent feature at patch $i$, codebook entry, and quantized feature at patch $i$, respectively. After that, VQGAN uses a decoder to reconstruct the image $\hat{x} = \text{Dec}(q)$. The training objective of VQGAN is to identify the optimal compression model of $\{\text{Enc}, \text{Dec}, \mathbf{C}\}$, involving the following loss:

$$\mathcal{L}_{\text{VQGAN}} = \mathcal{L}_{\text{rec}} + \mathcal{L}_{\text{VQ}} + \mathcal{L}_{\text{perceptual}} + \mathcal{L}_{\text{GAN}}, \tag{2}$$

where $\mathcal{L}_{\text{rec}}$ denotes the pixel reconstruction loss between $x$ and $\hat{x}$. $\mathcal{L}_{\text{VQ}}$ denotes the codebook loss that pulls the latent features $h$ and their closest codebook entries $q$ closer. $\mathcal{L}_{\text{perceptual}}$ denotes the perceptual loss between $x$ and $\hat{x}$ by leveraging a pre-trained vision model (Zhang et al., 2018). $\mathcal{L}_{\text{GAN}}$ introduces an adversarial training procedure with a patch-based discriminator (Isola et al., 2017) to calculate the GAN loss. We omit some detailed definitions for simplicity.

### 3.1 PILLAR 1: STRUCTURE DIVERSIFICATION VIA FACTORIZED QUANTIZATION

As shown in Fig. 2, we first improve the feature expressivity and representation capacity by diversifying the feature space. This is achieved by proposing a factorized quantization technique. It projects the image representation into several subspaces and performs quantization with independent sub-codebooks, each capturing different levels of semantics.

**Encoder.** We regard the original VQGAN encoder as a base feature extractor. On top of that, $k$ feature projectors are introduced to transform the base image features into their respective feature subspace. Formally,

$$h_{base} = \text{Enc}(x), \tag{3}$$

$$h_1, h_2, ..., h_k = F_1(h_{base}), F_2(h_{base}), ..., F_k(h_{base}), \tag{4}$$

where $F_1, ..., F_k$ are the projectors for each subspace branch.

**Quantizer.** Our method maintains a unique codebook for each semantic subspace. After projecting features into subspace, the quantization process is conducted at each codebook independently:

$$q_1, ..., q_k = \text{Quant}(h_1, \mathbf{C_1}), ..., \text{Quant}(h_k, \mathbf{C_k}), \tag{5}$$

where $\mathbf{C_1}, ..., \mathbf{C_k}$ are the corresponding sub-codebooks. After that, the visual token for each patch is obtained by aggregating the quantized features along the latent (channel) dimension:

$$q_{agg} = \text{Concat}([q_1; q_2; ...; q_k]). \tag{6}$$

**Decoder.** The aggregated features are then fed into the pixel decoder, which is inherited from the VQGAN model. Formally,

$$\hat{x} = \text{Dec}(q_{agg}). \tag{7}$$

**Why factorized quantization promotes representation learning?** First, the independent feature projection creates a basis for developing diverse feature space with different semantics. Second, the factorized lookup process greatly alleviates the lookup instability in a single codebook, allowing the existence of more latent codes that unlocks more expressive capacity. Lastly, the aggregation after quantization further extends the expressivity of the visual tokens, as it essentially builds an extremely large *conceptual* codebook with a size of $|\mathbf{C_i}|^k$.

### 3.2 PILLAR 2: EXPLICIT DISENTANGLEMENT

The factorized quantization design allows expressive and diverse feature learning, given the sufficient capacity brought by the independent feature projectors and sub-codebooks. However, without explicit constraints, the subspaces still risk learning redundant and overlapping features. To address this issue, we propose a disentanglement regularization mechanism for the decomposed subspace.

For simplicity, we take 2 subspaces as an example scenario. Through Eq. 5, we obtain $q_1 \in \mathbb{R}^{L \times D}$ and $q_2 \in \mathbb{R}^{L \times D}$, where $L$ is the number of patches. We design the disentanglement regularization mechanism as follows:

$$\mathcal{L}_{\text{disentangle}} = \frac{1}{n} \sum_{i=1}^{n} (q_1^\top q_2)^2, \tag{8}$$

where $n$ is the number of samples in a batch. This regularization mechanism minimizes the squared dot product between the two involved subspaces. The dot product directly measures the affinity between the two codes after L2 normalization, ranging from $[-1, 1]$, where -1/1 indicates negative/positive correlation and 0 denotes orthogonality. Minimizing the squaring function encourages the dot product value to approach 0. It also provides a smooth gradient for optimization. Note that this regularization does not directly apply to the sub-codebook. Instead, it operates on subspace features of each image instance. In other words, for each image, it encourages the involved each subspace to capture different aspects.

**Why disentanglement regularization is necessary?** The factorized quantization provides a foundation for rich semantics, but is not fully exploited, as the model is mainly driven the pixel reconstruction loss. Thus, the two subspace may risk convergering to the same feature distribution, both emphasizing the high-variance details. Disentanglement regularization explicitly penalizes redundancy by encouraging orthogonality between the subspace features, enforcing them to find different, complementary information to encode.

### 3.3 PILLAR 3: SEMANTIC FEATURE ALIGNMENT

Research (Balestriero & LeCun, 2024) suggests that the pixel reconstruction objective can hardly learn meaningful semantic features for perception, as the features mainly capture high-variance details. Recent works in visual generation (Yu et al., 2024e; Yao et al., 2025) demonstrate that it is beneficial to align the latent representations with semantic features. In this work, we adapt feature alignment into the factorized subspaces as an important pillar for enhancing representation.

Specifically, in the context of two subspaces, we only apply the feature alignment to one particular subspace to make it be specialized. In that case, all the factorized subspaces can collectively capture different levels of semantics. Concretely, one of the subspaces, say $C_2$, is tasked with predicting the features of a pre-trained vision model using a lightweight feature prediction model. $C_2$ essentially

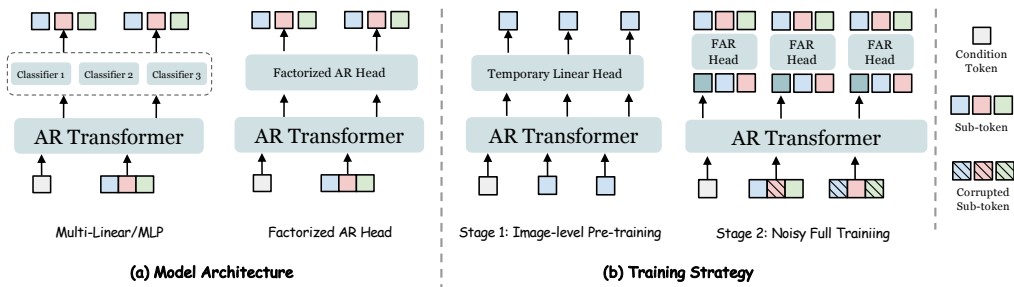

Figure 3: (a) Model architecture exploration on handling multi-token from multiple subspaces. (b) Staged training strategy with noisy sub-token enhances AR generation.

serves as the *semantic* subspace that embeds the semantic information. The other subspace $C_1$ functions as the *visual* one that captures the visual details, complementing $C_2$.

In visual generation, DINOv2 (Oquab et al., 2024) is the most widely adapted vision foundation models (VFMs) for providing semantic supervision. However, as studied in the multimodal domain (Tong et al., 2024), different VFMs place varying emphasis on the semantic property. For instance, CLIP (Radford et al., 2021) and DINOv2 (Oquab et al., 2024) are often regarded to capture high-level semantic and mid-level visual features respectively. In SSQ, we generalize the factorized quantization into *multiple* subspaces and incorporate diverse VFMs to establishes a hierarchy of semantics. For instance, an ideal triple-subspace would capture low-level structures (e.g., edges), mid-level details (e.g., textures), and high-level concepts (e.g., abstract appearance).

**Why feature alignment improves reconstruction?** Injecting semantic guidance is intuitively beneficial for enriching the semantic richness of the features space. Regarding reconstruction and generation, consider the example of an image patch depicting an ear. A traditional VQ code may capture its appearance, such as color, texture, etc. However, it is unaware of the species, *e.g.,* cat or dog. While such a code may effectively reconstruct the patch, introducing semantic information is expected to be beneficial. When informed with semantics, the decoder (and generation model) can better handle the corresponding visual reconstruction and generation tasks. Moreover, compared to high-variance signals, semantic information tends to generalize better.

**Synergy of Three Pillars.** The three pillars work in collaboration to promote the representation of visual tokens while improving reconstruction. The factorized quantization provides a *structure prior* in the feature spaces. Disentanglement teaches the model *how to factorize* by explicitly enforcing orthogonality. Feature alignment provides the "direction", guiding the model *where to decompose*. Reconstruction loss ensures high-quality reconstruction with whatever decomposition.

## 4 CO-DESIGN OF TOKENIZER & GENERATOR

While SSQ's design significantly enriches the semantic content of visual tokens for reconstruction, it introduces two fundamental challenges for standard auto-regressive (AR) generators. **1) An Architectural Mismatch**: SSQ produces a structured set of factorized sub-tokens for each image patch, whereas conventional AR models are designed to predict a single token per step. **2) A Performance Paradox**: Even after adapting the architecture, we find that improvements in reconstruction quality do not automatically transfer to generation. In fact, beyond a certain point, better reconstruction leads to worse generation—the Reconstruction-Generation Discrepancy.

This discrepancy was largely overlooked by prior work, which either evaluated dual-codebook tokenizers on specialized VAR models rather than standard AR generators (Li et al., 2024b; Qu et al., 2024) or did not scale their designs beyond two subspaces, where the effect is less pronounced. Resolving these challenges is not a matter of simple adaptation; it necessitates a fundamental co-design of the generator's architecture and training strategy.

### 4.1 ARCHITECTURE

We first explore how to update the standard AR model to handle multiple sub-codes for each patch. At the input side, we employ $k$ independent embedding layer plus an aggregation layer that com-

bines them into a single embedding. We compared this with vanilla adding strategy and found no significant difference. We keep this parametrized strategy for potential higher upper bound. In contrast, we found that the output head matters more, as shown in the left part of Fig. 3.

**Naive Baselines: Independent Classifiers.** A straightforward approach is to adapt the standard AR model's final linear layer. One could use multiple independent classifiers—either **Multi-Linear** heads or deeper **Multi-MLP** heads—with each one dedicated to predicting a sub-code for a specific subspace. However, both approaches fundamentally fail.

Their critical flaw is the implicit assumption that the sub-code predictions are conditionally independent given the transformer's output feature. This ignores the rich intra-patch dependencies that SSQ is designed to capture, leading the model to predict semantically incoherent combinations of sub-tokens. Empirically, these naive designs perform poorly, sometimes even worse than a baseline model using a simple, single-code tokenizer.

**Principled Solution: The Factorized AR Head.** To resolve this, we introduce a Factorized AR Head that explicitly models the dependencies between sub-codes within a single patch. Instead of making parallel, independent predictions, this head generates the sub-tokens sequentially, creating a "micro" auto-regressive model within the main prediction step. Formally, given the hidden state $g_t$ from the AR transformer backbone for a patch at timestep $t$, the head auto-regressively predicts the set of sub-tokens: $z_t^i = \text{head}_{AR}(g_t; z_t^1, z_t^2, \ldots, z_t^{i-1})$. This is implemented by feeding $g_t$ and the previously generated sub-token embeddings into a small auto-regressive module. Our experiments show this principled approach, which respects the structured nature of SSQ's output, outperforms the naive baselines and is critical for bridging the reconstruction-generation discrepancy.

## 4.2 TRAINING STRATEGY

While our co-designed architecture is effective for a dual-subspace tokenizer (SSQ-Dual), we observed that scaling to three subspaces (SSQ-Triple) causes a significant degradation in AR generation quality. This suggests that the increased complexity of the token space poses a severe optimization challenge. To address this, we introduce two complementary training strategies that are crucial for unlocking the full potential of our system.

**Staged Training** Our factorized AR model is tasked with learning a complex, hierarchical set of dependencies: 1) global, inter-patch relationships that define the overall image structure, and 2) local, intra-patch relationships between the disentangled sub-tokens. Training these two convoluted objectives simultaneously from scratch is inefficient and unstable, particularly as the number of sub-tokens increases. To de-couple and simplify this challenge, we propose a two-stage curriculum. **Stage 1: Pre-training for Global Structure**. We first train the main AR transformer backbone on simplified, single-code tokens. Using a standard linear head, the model's objective is solely focused on learning the high-level, inter-patch dependencies that are common to all images. This provides the model with a robust initialization and a strong understanding of global image composition. **Stage 2: Fine-tuning for Local Subspace Dependencies**. In the second stage, we introduce the full, multi-subspace SSQ tokens and integrate our factorized AR head. With the global dependencies already learned, the model can now efficiently focus its capacity on the more nuanced task of modeling the intricate relationships between the semantic sub-codes within each patch. This staged approach stabilizes training and leads to a more effective final model.

**Noisy Sub-token Training** A fundamental weakness of auto-regressive models is their susceptibility to cascading prediction errors. A single incorrect token prediction can corrupt the entire subsequent generation sequence. This problem is amplified in our setting, as the model must make multiple sub-token predictions per patch, increasing the risks for error. To mitigate this, we draw upon a key intuition: **the sub-tokens within a patch are semantically linked, and the overall patch representation should be robust to minor perturbations**. Based on this, we introduce a noisy sub-token training scheme. During training, for each patch, we randomly replace a fraction of its input sub-tokens with randomly sampled tokens from the codebook. However, the model is still tasked to predict the original, uncorrupted sub-tokens. This strategy acts as a powerful form of regularization, forcing the model to learn a more robust conditional distribution. By simulating the noisy conditions of actual inference—where its own previous predictions may be imperfect—the model becomes more resilient to error accumulation. This significantly improves the stability and quality of the final generated images.

Table 1: **Comparisons with other image tokenizers.** Reconstruction performance on $256 \times 256$ ImageNet. All models are trained on ImageNet, except "∗" on OpenImages.

| Method | DS Ratio | Codebook Size | rFID↓ | PSNR↑ |
|---|---|---|---|---|
| VQGAN (Esser et al., 2021) | 16 | 16384 | 4.98 | – |
| SD-VQGAN (Rombach et al., 2022) | 16 | 16384 | 5.15 | – |
| RQ-VAE (Lee et al., 2022) | 16 | 16384 | 3.20 | – |
| LlamaGen (Sun et al., 2024) | 16 | 16384 | 2.19 | 20.79 |
| Titok-B (Yu et al., 2024d) | – | 4096 | 1.70 | – |
| VQGAN-LC (Zhu et al., 2024b) | 16 | 100000 | 2.62 | 23.80 |
| VQ-KD (Wang et al., 2024b) | 16 | 8192 | 3.41 | - |
| VILA-U (Wu et al., 2024) | 16 | 16384 | 1.80 | - |
| Open-MAGVIT2 Luo et al. (2024) | 16 | 262144 | 1.17 | 21.90 |
| MergeVQ (Li et al., 2025) | 16 | 262144 | 1.12 | - |
| UniTok∗ (Ma et al., 2025) | 16 | 32768 | 0.33 | - |
| ImageFolder† (Li et al., 2024b) | 16 | 32768 | 0.80 | - |
| ImageFolder (Li et al., 2024b) | 16 | 32768 | 1.57 | - |
| TokenFlow (Qu et al., 2024) | 16 | 32768 | 1.37 | 21.41 |
| **SSQ-Dual** | 16 | 32768 | 0.94 | 22.02 |
| **SSQ-Triple** | 16 | 49152 | 0.76 | 22.73 |
| **SSQ-Quad.** | 16 | 65536 | 0.56 | 23.35 |
| SD-VAE† Rombach et al. (2022) | 8 | | 0.74 | 25.68 |
| SDXL-VAE† Podell et al. (2024) | 8 | – | 0.68 | 26.04 |
| ViT-VQGAN (Yu et al., 2022) | 8 | 8192 | 1.28 | – |
| VQGAN∗ (Esser et al., 2021) | 8 | 16384 | 1.19 | 23.38 |
| SD-VQGAN∗ (Rombach et al., 2022) | 8 | 16384 | 1.14 | – |
| OmniTokenizer (Wang et al., 2024a) | 8 | 8192 | 1.11 | – |
| LlamaGen (Sun et al., 2024) | 8 | 16384 | 0.59 | 25.45 |
| Open-MAGVIT2 (Luo et al., 2024) | 8 | 262144 | 0.34 | 26.19 |
| **SSQ-Dual** | 8 | 32768 | 0.32 | 26.27 |

Table 2: **Class-conditional generation on $256 \times 256$ ImageNet.** Models with the suffix "-re" use rejection sampling. The evaluation protocol and implementation follow ADM (Dhariwal & Nichol, 2021). SSQ inference adapts cfg, without top-k-top-p sampling.

| Type | Model | #Para. | FID↓ | IS↑ | Precision↑ | Recall↑ |
|---|---|---|---|---|---|---|
| Diffusion | ADM (Dhariwal & Nichol, 2021) | 554M | 10.94 | 101.0 | 0.69 | 0.63 |
| | CDM (Ho et al., 2022) | – | 4.88 | 158.7 | – | – |
| | LDM-4 (Rombach et al., 2022) | 400M | 3.60 | 247.7 | – | – |
| | DiT-XL/2 (Peebles & Xie, 2023) | 675M | 2.27 | 278.2 | 0.83 | 0.57 |
| | MAR (Li et al., 2024a) | 208M | 2.31 | 281.7 | 0.82 | 0.57 |
| LFQ AR | Open-MAGVIT2-B Luo et al. (2024) | 343M | 3.08 | 258.26 | 0.85 | 0.51 |
| | Open-MAGVIT2-L Luo et al. (2024) | 804M | 2.51 | 271.70 | 0.84 | 0.54 |
| VAR | VAR (Tian et al., 2024) | 310M | 3.3 | 274.4 | 0.84 | 0.51 |
| | ImageFolder Li et al. (2024b) | 362M | 2.60 | 295.0 | 0.75 | 0.63 |
| NAR | TiTok-S-128 (Yu et al., 2024d) | 177M | 1.97 | - | - | - |
| | MAGVIT-v2 (Yu et al., 2024b) | 307M | 1.78 | 319.4 | - | - |
| | MaskBit (Weber et al., 2024) | 305M | 1.62 | 338.7 | - | - |
| VQ AR | VQGAN (Esser et al., 2021) | 227M | 18.65 | 80.4 | 0.78 | 0.26 |
| | VQGAN (Esser et al., 2021) | 1.4B | 15.78 | 74.3 | – | – |
| | VQGAN-re (Esser et al., 2021) | 1.4B | 5.20 | 280.3 | – | – |
| | RQTran. (Lee et al., 2022) | 3.8B | 7.55 | 134.0 | – | – |
| | RQTran.-re (Lee et al., 2022) | 3.8B | 3.80 | 323.7 | – | – |
| | ViT-VQGAN (Yu et al., 2022) | 1.7B | 4.17 | 175.1 | – | – |
| | ViT-VQGAN-re (Yu et al., 2022) | 1.7B | 3.04 | 227.4 | – | – |
| | CRT-AR (Ramanujan et al., 2024) | 340M | 2.75 | 265.24 | 0.83 | 0.54 |
| | RAR-L (Yu et al., 2024c) | 461M | 1.70 | 299.5 | 0.81 | 0.60 |
| | RandAR (Pang et al., 2025) | 343M | 2.55 | 288.82 | 0.81 | 0.58 |
| | IAR (Hu et al., 2025) | 343M | 3.18 | 234.8 | 0.82 | 0.53 |
| | LlamaGen-L (Sun et al., 2024) | 343M | 3.80 | 248.28 | 0.83 | 0.51 |
| | LlamaGen-XL (Sun et al., 2024) | 775M | 3.39 | 227.08 | 0.81 | 0.54 |
| | **SSQ-LlamaGen-L** | 415M | 2.61 | 313.88 | 0.78 | 0.57 |

# 5 EXPERIMENT

**Model Setup** The SSQ tokenizer and AR generation model are modified based on the implementation of LlamaGen (Sun et al., 2024). The tokenizers are trained with a global batch size of 256 and a constant learning rate of 2e-4 across 8 A100 GPUs. The AR models are trained with a constant learning rate of 2e-4 and a global batch size of 256 across 8 A100 GPUs. When evaluating generation, we use classifier-free guidance (Ho & Salimans, 2022) (CFG). We do not use any top-k or top-p sampling strategy unless specified.

**Dataset and Metric** We follow previous works to use ImageNet (Deng et al., 2009) for training and evaluation. We adopt Fréchet inception distance (FID) (Heusel et al., 2017) as the main metric. For tokenizers, we use the ImageNet validation set, consisting of 50k samples, to compute the reconstruction FID (rFID). For generation models, we employ the ADM (Dhariwal & Nichol, 2021) evaluation protocol to compute the generation FID (gFID). PSNR, Inception Score, Precision, and Recall are also reported for reference.

## 5.1 MAIN RESULTS

**Comparison on Tokenizers** We compare SSQ with popular visual tokenizers in Tab. 1. SSQ sets a new state-of-the-art performance in discrete image reconstruction across various settings. Compared to VQGAN and its advanced variants, our method outperforms them by a large margin. As our implementation is based on LlamaGen (Sun et al., 2024), the performance gap between LlamaGen and SSQ clearly demonstrates the effectiveness of the semantic subspace quantization design.

Comparing with ImageFolder (Li et al., 2024b)[1] and TokenFlow (Qu et al., 2024) that also use separated codebooks, SSQ also shows better performance. We attribute the difference to that the explicit disentanglement benefits learning a semantically expressive feature space, while the multi-source feature alignment further improves the capacity and diversity.

Notably, SSQ demonstrates decent scalability in reconstruction. As shown in Tab. 1, increasing the codebook size (i.e., number of codebooks) leads to consistent performance gain in reconstruction. SSQ-Quad in $16\times$ downsample ratio is even better than LlamaGen in $8\times$ downsample ratio.

**Comparison on Generation Models** We apply SSQ on the standard AR model and compare it with mainstream image generation models, as shown in Tab. 2. Among VQ-based AR models, we observe that our model achieves competitive image generation performance. When comparing to

---

[1]ImageFolder† uses a specially tailored DINO discriminator to boost the performance, we ablate this component to better illustrate the comparison on the core method on shaping semantic space.

models with similar parameter sizes and architectures, specifically SSQ-LlamaGen vs. LlamaGen-L, our model achieves significantly superior performance in both FID (2.61 vs. 3.80) and Inception Score (313.88 vs. 248.28). This gap clearly validates the effectiveness of the proposed method.

## 5.2 Understanding Semantic Subspace Quantization

**Necessity of the three pillars.** We conduct a series of ablation studies in Tab. 3 to dissect the contribution of each of our three core principles. We begin by confirming the limitations of the conventional approach. As shown in Tab. 3, naively increasing a monolithic codebook's size not only fails to improve performance but actually degrades reconstruction quality (rFID increases). This validates our premise that simply scaling the codebook is a flawed strategy.

Table 3: Ablation study on the necessity of each mechanism in SSQ. Tokenizers trained for 10 epochs and generators trained for 200 epochs.

| Model | #Codes | #Subspaces | rFID | IS | gFID |
|---|---|---|---|---|---|
| Baseline | 16384 | 1 | 3.71 | 50.05 | 4.85 |
| Baseline | 32768 | 1 | 3.60 | 50.60 | - |
| +Structure Diverse. | 32768 | 2 | 2.00 | 54.72 | 4.03 |
| +Explicit Disentangle. | 32768 | 2 | 1.84 | 55.04 | 3.88 |
| +Feature Alignment | 32768 | 2 | 1.66 | 55.21 | 3.66 |
| +Multi-source Align. | 49152 | 3 | 1.30 | 56.41 | 4.18 |

In stark contrast, introducing our first principle, Structural Diversification, yields a dramatic improvement. By decomposing the single large codebook into two smaller, factorized subspaces, the reconstruction rFID plummets from 3.71 to 2.00. This powerful result underscores that the structure of the latent space is far more critical than its raw size.

Building on this strong, structurally diversified baseline, we then incrementally integrate our other two principles. Adding the Explicit Disentanglement further improves reconstruction, and the Feature Alignment pushes the performance even lower. Notably, at this dual-codebook scale (SSQ-Dual), these consistent gains in reconstruction successfully translate to improved downstream generation, as evidenced by the corresponding drop in gFID.

We push the framework's limits by scaling to a triple-subspace design (SSQ-Triple). The results are telling: reconstruction quality continues to improve, achieving an even lower rFID. However, this pursuit of reconstruction perfection reveals the central paradox. While the rFID continues to decrease, the generation performance (gFID) not only stops improving but begins to degrade significantly. This is the first empirical evidence of the Reconstruction-Generation Discrepancy. The design choices that create a richer, more complex token space for reconstruction simultaneously violate the implicit assumptions of standard AR models, creating a "pain" for generation. This finding unequivocally demonstrates that a superior tokenizer is not a plug-and-play solution and highlights the critical need for the tokenizer-generator co-design we introduce next.

**In-depth Analysis of SSQ Feature Space** To understand how SSQ achieves its performance, we conducted an in-depth analysis of its learned feature space. Through a series of qualitative and quantitative experiments, we verify that our framework successfully produces specialized and provably disentangled subspaces that learn complementary visual roles (e.g., separating color from structure). Furthermore, t-SNE visualizations reveal that semantic guidance imposes a meaningful, class-based topology on the latent codebooks, with the structure reflecting the distinct properties of the guide models. Due to space constraints, the full investigation, complete with visualizations and detailed metrics, is provided in Appendix.

Table 4: Comparison of AR prediction head.

| Architecture | gFID↓ | IS↑ |
|---|---|---|
| Multi-Linear | 5.19 | 175.10 |
| Multi-MLP | 5.59 | 164.53 |
| AR Head | 4.37 | 244.02 |

## 5.3 Co-Design of Tokenizer & Generator

**Co-Design Model Architecture** While SSQ excels at reconstruction, its factorized token structure presents a new architectural challenge for AR generators, which must now predict a structured set of sub-codes per patch rather than a single token. Tab. 4 evaluates our design choices on model architecture. We find that naive approaches, such as using independent linear or MLP classifiers for each subspace, are suboptimal. These methods fail because they ignore the crucial intra-patch

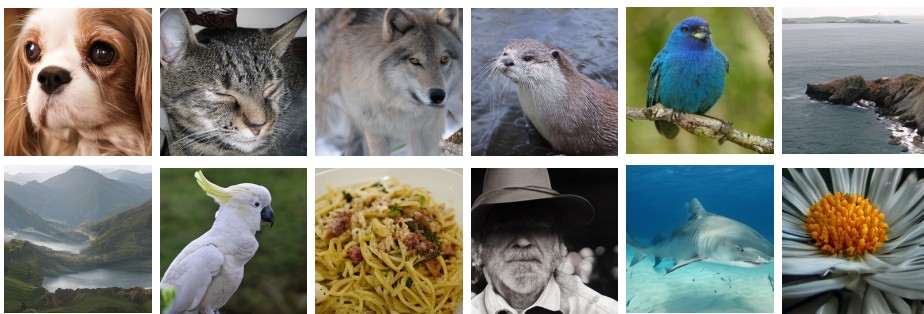

Figure 4: Qualitative results of visual generation.

dependencies, treating each sub-code prediction as an isolated event. This can lead to incoherent combinations of semantics within a single image patch.

In contrast, our factorized AR head explicitly models these dependencies. By predicting the sub-codes for a patch sequentially, it conditions each prediction on the previously generated sub-codes for that same patch. As the results show, this design is superior because it creates a more holistic, two-level auto-regressive process: the AR backbone models the global dependencies between patches, while the AR head models the local dependencies within each patch's set of sub-codes.

**Co-Design Training Strategy**   While our architectural co-design is effective for SSQ-Dual, the more powerful SSQ-Triple presents a significant optimization challenge. As shown in Tab. 5, a standard training approach fails to harness SSQ-Triple's superior representational capacity; it paradoxically performs worse than the baseline model, yielding a negative $\Delta gFID$. This underscores the need for a co-designed training strategy.

Our proposed strategies systematically resolve this issue. First, the staged training curriculum is crucial. By decomposing the learning of global (inter-patch) and local (intra-patch) dependencies, it stabilizes optimization and allows SSQ-Triple to finally outperform the baseline, turning the $\Delta gFID$ positive. Next, the noisy sub-token strategy provides another substantial gain. This technique acts as a powerful regularizer that improves the model's robust-

Table 5: Comparison on the training strategies of the AR model.

| Training Strategy | Tok. | gFID↓ | IS↑ | $\Delta$gFID↑ |
|---|---|---|---|---|
| Baseline | Single | 3.80 | 248.28 | - |
| SSQ Vanilla | Triple | 3.86 | 259.21 | -0.06 |
| +Staged Training | Triple | 3.56 | 275.12 | 0.24 |
| +Noisy Sub-token | Triple | 3.31 | 286.90 | 0.49 |

ness to the cascading prediction errors inherent in auto-regressive modeling. By training the model to be resilient to its own potential mistakes during inference, it further widens the performance gap in our favor.

Together, these co-designed strategies are essential, transforming the underperforming SSQ-Triple into a model that decisively surpasses the baseline and validates our holistic approach. More experimental results and investigations are provided in the Appendix.

## 6   CONCLUSION

In this work, we addressed the fundamental representation trade-off limiting discrete visual tokenizers. We began by proposing a principled framework for enhancing token expressivity, built on the core pillars of feature alignment, structural diversification, and explicit disentanglement. Our tokenizer, SSQ, materializes this framework to achieve state-of-the-art reconstruction performance. Our another key contribution was the discovery of a critical paradox: the very design choices that perfect reconstruction can catastrophically degrade the performance of standard auto-regressive generators. We identified this as the Reconstruction-Generation Discrepancy, a fundamental tension previously overlooked in the field. To resolve this, we introduced a novel tokenizer-generator co-design philosophy. By tailoring both the generator's architecture and its training strategy to the rich structure of our advanced tokens, we successfully alleviated this issue. We hope this work could inspire more future work on tokenizer-oriented generator design that can better utilize the power of tokenizer on visual generation.

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

# A APPENDIX

## A.1 LLM USAGE

In preparing this manuscript, we used Gemini-2.5-Pro, solely for language polishing and minor refinements to improve clarity and flow in the text. The LLM was provided with sections written by the authors and instructed to suggest revisions, which were then reviewed, edited, and incorporated by the authors as deemed appropriate. All core ideas, research contribution, technical details, and analysis originate from the authors and were not generated or ideated by the LLM. No other LLMs were used in the research process.

## A.2 IN-DEPTH ANALYSIS OF FEATURE SPACE OF SSQ

Having established the empirical success of SSQ, we now dissect its internal mechanics to understand *what* its structured subspaces learn. We conduct a series of analyses to visualize their functional roles, quantify their properties, and map their semantic topology.

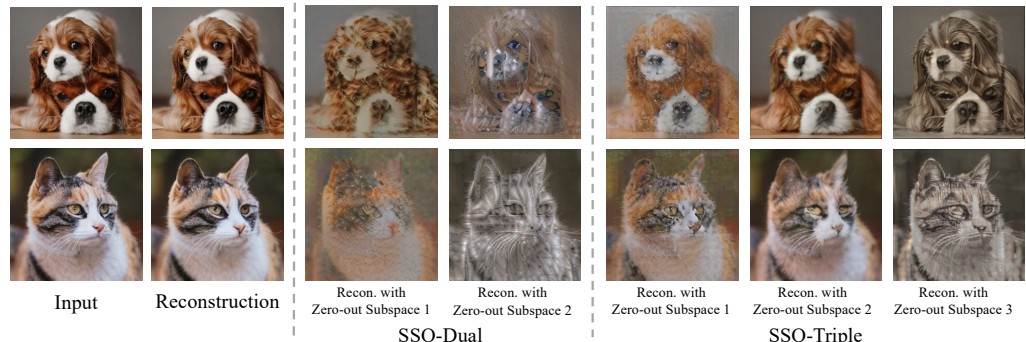

Figure 5: Visualization of standard reconstruction by SSQ and reconstruction by zeroing-out one subspace.

**Qualitative Analysis: What Does Each Subspace Learn?** To qualitatively understand the specialization of each subspace, we visualize reconstructions where one subspace is selectively deactivated by "zeroing it out" (Fig. 5). For SSQ-Dual, the roles are distinct: zeroing out Subspace 2 removes most color information, leaving Subspace 1 to render a structural, textural backbone.

For SSQ-Triple, this specialization becomes even more nuanced. Deactivating Subspace 3 results in a near-complete loss of color, confirming its role as the primary chrominance channel. The remaining two subspaces appear to handle structure at different frequencies, with Subspace 1 capturing high-frequency textures (e.g., overall shape) and Subspace 2 managing low-frequency components (e.g., fur). This provides clear visual evidence that the subspaces learn disentangled, complementary visual attributes.

**Quantitative Validation: Verifying Guidance and Disentanglement** We now quantitatively validate these observations. As shown in Fig. 6(a), we compute the feature affinity between each learned subspace and the vision foundation models (VFMs) used for guidance. The results are unequivocal: Subspaces 2 and 3 exhibit high similarity to their respective guides (DINOv2 and CLIP), while the reconstruction-focused Subspace 1 remains neutral. This not only proves the effectiveness of our feature alignment mechanism but also explains the functional specializations observed above; for instance, the CLIP-guided subspace consistently learns to represent color information across both SSQ-Dual and SSQ-Triple models.

Furthermore, we validate our disentanglement objective by computing the dot product between the feature subspaces themselves (Fig. 6(b)). The resulting matrix is strongly diagonal, with near-zero off-diagonal values. This provides powerful evidence that our regularization is successful, enforcing orthogonality and ensuring that the subspaces learn non-overlapping representations.

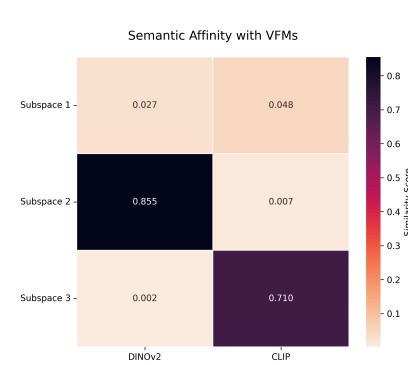 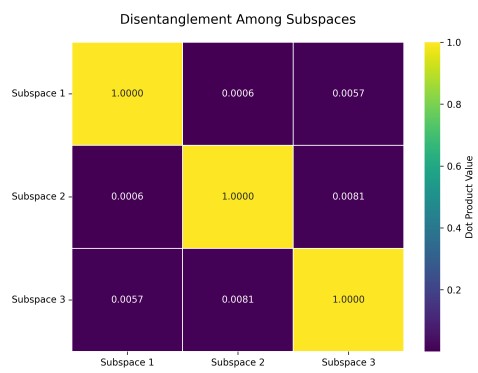

(a) Affinity between feature of SSQ and VFM.      (b) Affinity among subspaces of SSQ.

Figure 6: Feature subspaces affinity. (a) Subspaces 2 and 3 show high similarity to DINOv2 and CLIP features, respectively, demonstrating feature alignment. (b) The dot product between subspaces is near-zero off-diagonal, indicating disentanglement.

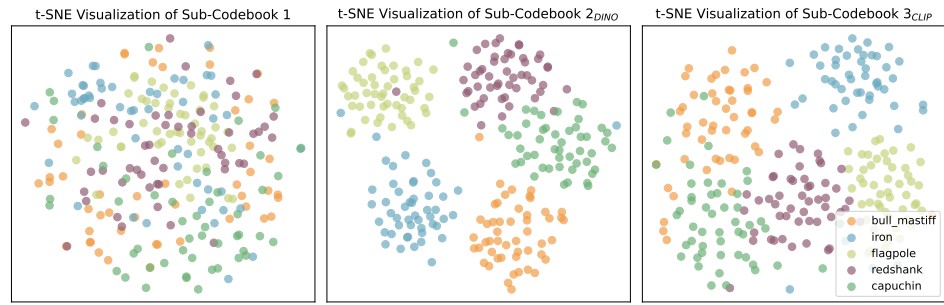

Figure 7: T-SNE visualization of VQ codes from different subspace in SSQ-Triple.

**Latent Space Topology** Finally, we investigate how SSQ organizes the topology of the codebook embeddings. We use t-SNE to visualize the code embeddings from SSQ-Triple after encoding images from five ImageNet classes (Fig. 7). The resulting distributions are highly revealing:

- Subspace 1 (Reconstruction-focused): The embeddings are scattered and unstructured, consistent with a subspace dedicated to capturing high-variance, low-level image details.
- Subspaces 2 & 3 (Semantically-guided): In stark contrast, the embeddings form distinct, well-separated clusters corresponding to the image categories. This demonstrates that the representation learning objective successfully imposes a semantically meaningful structure on the latent space.

Interestingly, the DINOv2-guided Subspace 2 forms more compact, tightly-defined clusters than the CLIP-guided Subspace 3. We attribute this to the differing strengths of the guide models; the DINOv2 model used is a superior ImageNet classifier (81.1% accuracy) compared to the CLIP model (68.3%). This performance gap is mirrored in the latent space topology, where the stronger vision-centric classifier (DINOv2) induces a more class-separable organization. This suggests SSQ naturally forms a semantic hierarchy, where different guides impart distinct and predictable structural properties on the latent space.

## A.3    THE PERSISTENT GAP BETWEEN RECONSTRUCTION AND GENERATION

Our co-design methodology proves to be universally effective, systematically improving generative performance for both the SSQ-Dual and the more complex SSQ-Triple models, as detailed in Tab. 6.

However, this analysis also uncovers a subtle yet persistent challenge. A performance gap remains between the two models, where SSQ-Triple, despite its superior reconstruction capability, still lags behind SSQ-Dual in generation. While each of our co-design strategies successfully narrows this gap—progressively reducing the $\Delta gFID$—they do not eliminate it entirely.

Table 6: Comparison on the training strategies of the AR model.

| Training Strategy | Tok. | gFID↓ | IS↑ | ΔgFID↓ |
|---|---|---|---|---|
| LlamaGen Baseline | Single | 3.80 | 248.28 | - |
| SSQ Vanilla | Dual | 3.38 | 248.26 | 0.48 |
| | Triple | 3.86 | 259.21 | |
| +Staged Training | Dual | 3.17 | 260.36 | 0.39 |
| | Triple | 3.56 | 275.12 | |
| +Noisy Sub-token | Dual | 3.11 | 260.83 | 0.20 |
| | Triple | 3.31 | 286.90 | |

This finding points to a deeper, unresolved question in the field. Ideally, a tokenizer with superior representational power (evidenced by better reconstruction) should yield superior generative performance. The remaining gap suggests that fully translating reconstruction advancements into generative excellence remains a fundamental challenge, presenting a fertile ground for future research into even more sophisticated tokenizer-generator co-design.

### A.4 LIMITATION

Despite the promising results, this work presents several limitations that open avenues for future investigation. First and foremost, while our proposed co-design methodology significantly narrows the Reconstruction-Generation Discrepancy, a residual performance gap persists, particularly when scaling from a dual-subspace to a triple-subspace tokenizer (SSQ-Dual vs. SSQ-Triple). As shown in our experiments, the model with superior reconstruction capabilities (SSQ-Triple) does not yet achieve superior generation performance, indicating that the optimization challenge is not fully resolved. This underscores the core thesis of our work and highlights a critical area for future research: developing even more sophisticated generator architectures and training paradigms that can fully leverage the rich, structured information from advanced tokenizers.

Furthermore, the scalability of our approach beyond three or four subspaces remains an open question. While reconstruction quality continues to improve, it is likely that both the optimization challenges for the generator and the computational complexity of the factorized AR head will intensify. Future work could explore more efficient co-design strategies to mitigate these costs.

Finally, our experiments are primarily focused on class-conditional image generation on ImageNet. The generalizability of the SSQ framework and the co-design principles to other domains, such as text-to-image generation, or other architectures, like latent diffusion models, warrants further exploration. We believe that investigating these challenges will continue to advance the field, reinforcing the necessity of a synergistic approach to tokenizer and generator design.

## B MORE GENERATION RESULTS

Figure 8 presents additional examples generated by our SSQ model, highlighting its impressive image generation capabilities.

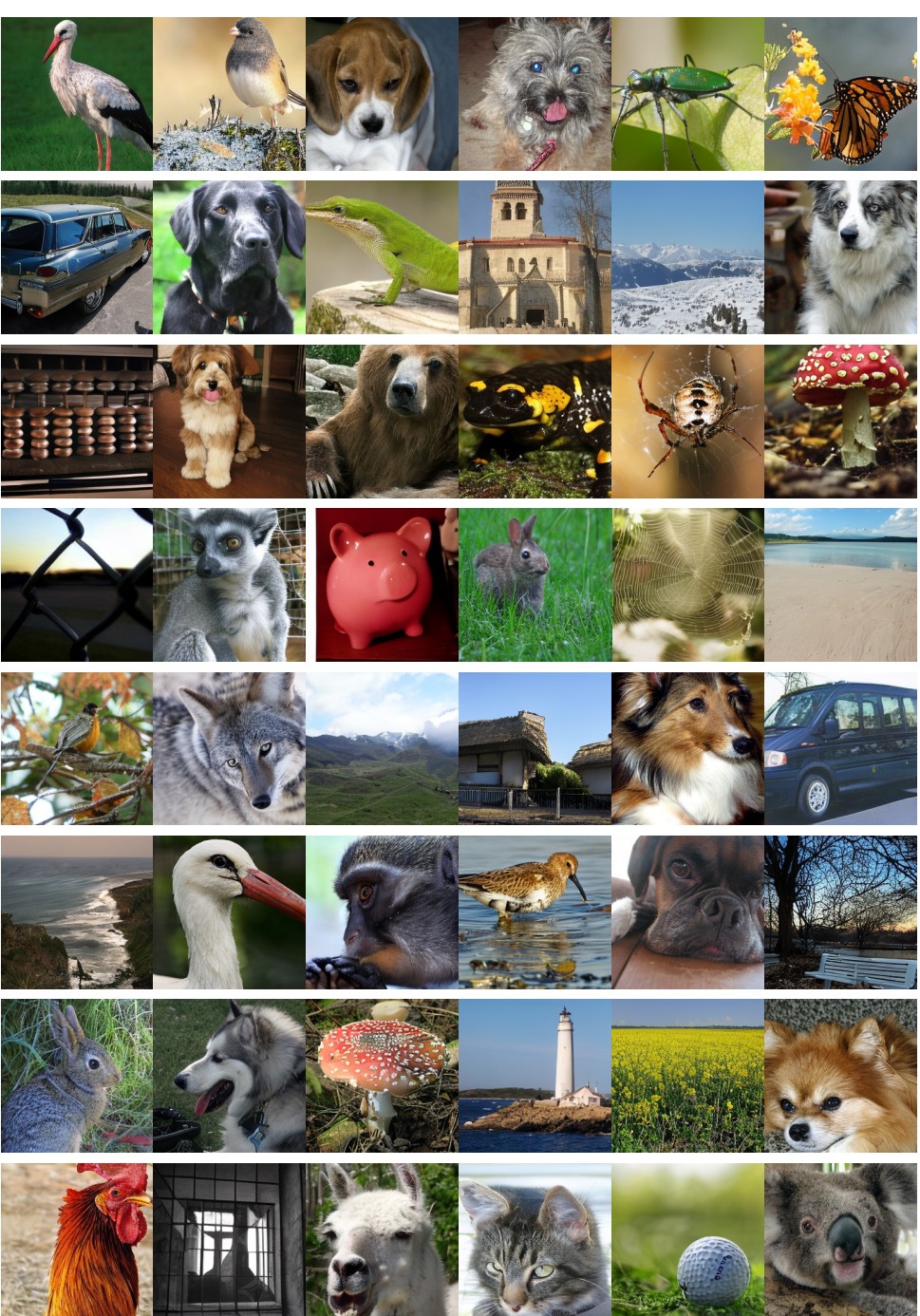

Figure 8: More qualitative examples generated by the SSQ AR model.

