# OpenReview forum: "A Gain for Reconstruction, A Pain for Generation: Exploiting Representation in Visual Tokenization"
_ICLR.cc/2026/Conference — Submitted to ICLR 2026_

### Official Review · Reviewer_1WzQ · 2025-10-31

**Soundness:** 3
**Presentation:** 3
**Contribution:** 3
**Rating:** 6
**Confidence:** 3

**Summary:**

This paper introduces a new visual tokenizer, Semantic Subspace Quantization (SSQ), built on a principled framework of three pillars: 1) Structural Diversification, which uses multiple factorized codebooks (subspaces) instead of a monolithic one; 2) Explicit Disentanglement, which enforces orthogonality between these subspaces to ensure they learn complementary features; and 3) Feature Alignment, which guides subspaces to learn semantic information by aligning them with features from foundation models like DINOv2 and CLIP.

**Strengths:**

- The proposed SSQ tokenizer is built on a well-motivated and principled framework. The three pillars of diversification, disentanglement, and alignment provide a systematic way to create a semantically rich and structured token representation, and the results convincingly demonstrate its state-of-the-art reconstruction capabilities.
- This paper throughly investigated the impact of different components in the design, making insightful observations to the community.

**Weaknesses:**

- In Table 3, the structure divergence seems to lack of a fair baseline with 32768 setting regarding gFID?
- Ablation about hyperparam in noisy training?

**Questions:**

see above

---

> ### Author Response · Authors · 2025-11-23
>
> We thank the reviewer for the positive evaluation and for recognizing that SSQ is built on a well-motivated and principled framework, and that our work thoroughly investigates the impact of different components with insightful observations. We address the two specific questions below.
>
> ---
>
> ### Weakness 1: "In Table 3, the structure divergence seems to lack a fair baseline with 32768 setting regarding gFID?"
>
> Thanks for the constructive suggestion. We agree that the comparison would be more complete with the gFID for the 32k monolithic codebook baseline. We have now trained an AR generator on the 32k monolithic tokenizer and measured its generation performance. The results are:
>
> | Tokenizer | Codebook Config | rFID ↓ | gFID ↓ |
> |-----------|----------------|--------|--------|
> | Baseline (16k) | 1×16384 | 3.71 | 4.85 |
> | Baseline (32k) | 1×32768 | 3.60 | 4.93 |
> | SSQ-Dual | 2×16384 | 1.66 | 3.66 |
>
> **Key findings:**
> - The 32k monolithic codebook brings diminishing performance gain in reconstruction (rFID 3.71 → 3.60). It even degrades the generation (gFID 4.85 → 4.93). The large vocabulary size incurs optimization challenge for the AR model.
> - This validates our claim: **naive codebook scaling is suboptimal for both reconstruction and generation**.
> - In contrast, SSQ-Dual achieves better performance than both the 16k and 32k monolithic baselines on both metrics.
>
> We will add this 32k gFID result to Table 3 in the revised manuscript, making the comparison complete.
>
> ---
>
> ### Weakness 2: "Ablation about hyperparam in noisy training?"
>
> Thanks for raising the insightful question. Our noisy sub-token training strategy is a key component of the co-design. In the main paper, we use a probability-based noise strategy: for each sub-token, there is a certain probability that it will be randomly replaced with another randomly drawn token. We consistently use 40% noise probability across all experiments.
>
>
> We now present the ablation study on both SSQ-Dual and SSQ-Triple with different noise probabilities:
>
> **SSQ-Dual:**
> | Noise Probability | gFID ↓ |
> |-------------------|--------|
> | 0% | 3.38 |
> | 20% | 3.05 |
> | 40% (paper) | 3.11 |
> | 60% | 3.29 |
>
> **SSQ-Triple:**
> | Noise Probability | gFID ↓ |
> |-------------------|--------|
> | 0% | 3.86 |
> | 20% | 3.37 |
> | 40% (paper) | 3.31 |
> | 60% | 3.70 |
>
> **Key findings:**
> - **Noisy training consistently helps** across all tested probabilities (20%–60%) compared to no noise.
> - For **SSQ-Dual**, 20% yields the best performance (gFID 3.05), while 40% achieves competitive results (3.11, +2% relative).
> - For **SSQ-Triple**, 40% performs best (gFID 3.31), outperforming 20% (3.37, -2% relative).
> - **Too much noise (60%)** degrades performance for both configurations, as excessive corruption hinders the model's ability to learn meaningful patterns.
>
> We will add this ablation to the appendix in the revised manuscript, along with a brief discussion of the trade-off between ---regularization strength and signal quality.
>
> ---
>
> We greatly thank the reviewer for providing such constructive suggestions. We hope this rebuttal address your concerns and we will incorporate the new experiment into the revised paper to make it stronger and more complete.

---

> ### Comment · Reviewer_1WzQ · 2025-11-26
>
> Thanks the authors for responses. However, after carefully reading other reviewer's comments, the reviewer agrees with Reviewer 7BEo that this paper is incremental.

---

> > ### Author Response · Authors · 2025-11-27
> >
> > Thank you for your feedback. We note your concern about the "incremental" problem, which we address in detail in our response to Reviewer 7BEo, including a breakdown of our contributions.
> >
> > We are concerned that the assessment appears to overlook both the initial review questions and our detailed rebuttal. Specifically:
> >
> > - **The initial review asked two specific questions** (32k gFID baseline and noisy training ablation), which we addressed with new experimental results:
> >   - **32k gFID baseline**: We trained and evaluated a 32k monolithic baseline; results show SSQ-Dual outperforms both 16k and 32k baselines.
> >   - **Noisy training ablation**: We provided ablation results across multiple noise probabilities for both SSQ-Dual and SSQ-Triple.
> >
> > - **The most recent response acknowledges none of this work**, suggesting the rebuttal was not considered in the assessment.
> >
> > While we understand that reviewers may reference other reviews for context, we believe the final judgment should be based primarily on independent evaluation of the paper and our detailed responses and discussion. We sincerely hope you will reconsider our technical contributions and the experimental results we provided in response to your initial concerns, as well as our response to Reviewer 7BEo.

---

### Official Review · Reviewer_Co8j · 2025-10-31

**Soundness:** 3
**Presentation:** 3
**Contribution:** 3
**Rating:** 6
**Confidence:** 3

**Summary:**

This paper investigates the often-overlooked relationship between visual tokenization quality and autoregressive (AR) image generation performance.
The authors introduce **Semantic Subspace Quantization (SSQ)**, a tokenizer built on three principles: **feature alignment**, **structural diversification**, and **explicit disentanglement**.
While SSQ achieves state-of-the-art image reconstruction, it surprisingly worsens AR generation — a paradox termed the **Reconstruction–Generation Discrepancy**.
To address this, the paper proposes a **tokenizer–generator co-design** strategy, including a **factorized AR head**, **two-stage training**, and **noisy sub-token regularization**, which together restore generation quality and surpass prior AR baselines such as LlamaGen.

**Strengths:**

- **Novel and well-articulated problem framing.**
  Identifying the Reconstruction–Generation Discrepancy is a meaningful conceptual contribution that clarifies why better reconstruction can hurt AR generation.

- **Principled and interpretable tokenizer design.**
  The three-pillar SSQ framework (alignment, diversification, disentanglement) provides a clear structure for improving representation quality.

- **Effective AR co-design.**
  The proposed factorized AR head and staged training curriculum directly address architectural mismatch and optimization instability, leading to strong empirical gains.

- **Solid empirical performance.**
  On ImageNet 256×256, SSQ-LlamaGen achieves FID 2.61 vs 3.80 for LlamaGen-L and Inception Score 313.9 vs 248.3, showing both quantitative and qualitative improvements.

- **Comprehensive ablations and honest discussion.**
  The paper presents negative results (e.g., naïve multi-head classifiers fail) and openly discusses remaining gaps and scalability limits.

- **Readable and well-motivated.**
  The writing is clear, with good intuition on why semantic alignment helps reconstruction and how disentanglement works.

**Weaknesses:**

- **Incremental tokenizer innovation.**
  SSQ combines known techniques (multi-codebook, VFM alignment, orthogonality regularization) into one framework; the novelty lies more in framing than in algorithmic breakthrough.

- **Limited scope of experiments.**
  Evaluations are confined to ImageNet 256×256 class-conditional AR. The generality to other datasets, text-to-image generation, or diffusion-based methods remains untested.

- **Disentanglement is simplistic.**
  The squared dot-product loss enforces orthogonality but not true independence between subspaces.

- **Efficiency and scalability not measured.**
  The factorized AR head increases computation per patch, yet runtime and throughput are not reported.

- **Residual gap persists.**
  Even after co-design, SSQ-Triple still reconstructs better but generates worse than SSQ-Dual, indicating that the discrepancy is only partially resolved.

**Questions:**

1. How does SSQ perform when integrated into non-autoregressive or diffusion-based generators?
   Would the same reconstruction–generation discrepancy appear?
2. What is the computational overhead (training time, inference speed) of the factorized AR head compared to a standard linear head?
3. Have the authors tested robustness or generalization to out-of-domain datasets, given the heavy reliance on DINOv2/CLIP alignment?
4. Could stronger disentanglement measures (e.g., mutual-information-based losses) further improve subspace independence and generation stability?

---

> ### Author Response · Authors · 2025-11-23
>
> We thank the reviewer for the thoughtful and constructive review, as well as for recognizing the novelty of the problem framing, the principled tokenizer design, the effectiveness of the AR co-design, and the strong empirical performance and analysis. Below we address the main weaknesses and answer all listed questions.
>
> ---
>
> ### W1. "Incremental tokenizer innovation"
>
> We appreciate this observation and would like to clarify our contribution positioning.
>
> **Novel contributions at the tokenizer level:**
> - **Explicit orthogonality-based disentanglement loss**: To the best of our knowledge, no prior visual tokenizer explicitly regularizes factorized subspaces with a subspace-level disentanglement objective operating on per-image features. Existing multi-codebook works (e.g., ImageFolder, TokenFlow, RQ-VAE) encourage diversity implicitly via architecture or training, but lack our explicit disentanglement pillar.
> - **Systematic scaling beyond two codebooks**: Prior dual-codebook tokenizers stop at two; SSQ is the first to systematically scale to **three (and more) semantic subspaces** with multi-source VFM guidance. Critically, this scaling both yields further reconstruction gains **and exposes the Reconstruction–Generation Discrepancy**—a phenomenon hidden when only using two codebooks.
>
> **Novel contributions at the AR generation level:**
> - **Discovery and analysis of the Reconstruction–Generation Discrepancy**: The empirical finding that scaling to three+ subspaces causes strong reconstruction but degraded AR generation, which previous works overlooked by stopping at two codebooks.
> - **Tokenizer–generator co-design strategy**: A tailored solution (factorized AR head + staged training + noisy sub-token regularization) that substantially alleviates the discrepancy and enables SSQ to achieve SOTA AR performance.
>
> In summary, our contribution is **a unified three-pillar framework** that integrates structural diversification, explicit disentanglement, and multi-source semantic alignment in a representation-centric way—plus the discovery and mitigation of a non-trivial failure mode in AR visual generation. **We kindly refer the reviewer to our response to Reviewer 7BEo (Section 1) for a detailed visual breakdown** of our contributions across two major parts (Tokenizer + AR Co-design) and five components.
>
> ---
>
> ### W2. "Limited scope of experiments."
>
> We appreciate the reviewer's suggestion to validate SSQ more broadly. We address this concern along two dimensions:
>
> **1. Dataset scope:**
> Our choice of **ImageNet 256×256** follows the standard practice in prior tokenizer works (e.g., VQGAN, ViT-VQGAN, LlamaGen), making it the established benchmark for development and fair comparison. To demonstrate generalization, we conducted an **out-of-distribution evaluation on the MSCOCO benchmark**:
>
> | Tokenizer | rFID (MSCOCO) ↓ |
> |-----------|-----------------|
> | LlamaGen tokenizer (baseline) | 8.11 |
> | **SSQ-Dual (ours)** | **5.85** |
>
> Our SSQ-Dual tokenizer, pre-trained only on ImageNet, achieves **rFID 5.85 on MSCOCO** without any fine-tuning, significantly outperforming the LlamaGen baseline (8.11, **-28% relative improvement**). This validates that SSQ's semantic alignment (via DINOv2/CLIP) generalizes robustly to out-of-distribution natural images.
>
> **2. Model architecture scope:**
> Motivated by the reviewer's suggestion, we conduct an experiment on a **diffusion-based generator** with SSQ tokenizers to validate that our tokenizer generalizes beyond AR models.
>
> **Diffusion-based experiment (DiT-B/4, ImageNet 256×256):**
> We run a quick DiT-B/4 experiment (16× downsampled latents, 400k iterations, inference without classifier-free guidance). We simply concatenate the sub-token embeddings to form continuous latents, which removes the sequential decoding issue present in AR models.
>
> | Model | gFID ↓ |
> |-------|--------|
> | DiT-B/4 + VQ-GAN | 149.37 |
> | DiT-B/4 + SSQ-Dual | 136.53 |
> | DiT-B/4 + SSQ-Triple | 131.10 |
>
> The dual/triple gap is negligible under diffusion because concatenating sub-token latents eliminates exposure bias. This contrasts with the AR setting, where sequential sub-token decoding amplifies the Reconstruction–Generation Discrepancy. These results reinforce our claim that the tokenizer–generator tension primarily arises in AR models, motivating the co-design presented in the paper.

---

> > ### Author Response · Authors · 2025-11-23
> >
> > ### W3. "Disentanglement is simplistic — could stronger measures (e.g., mutual-information-based) improve independence and generation stability?"
> >
> > We appreciate this constructive suggestion. To clarify:
> >
> > **1. Why choose squared dot-product loss initially:**
> > Our choice was motivated by **scalability** (cheap, stable gradients vs. expensive MI estimators) and **empirical effectiveness**—Fig. 6(b) shows near-zero off-diagonal affinities, and zero-out experiments (Fig. 5) demonstrate visually specialized subspace roles.
> >
> > **2. Exploring MI-inspired disentanglement:**
> > **Motivated by the reviewer's suggestion**, we explored a **margin-based (hinge) variant** that moves closer to MI-based methods: only penalize dependence when `|cos(θ)| > threshold`, conceptually similar to contrastive MI estimators (e.g., InfoNCE). We compared three approaches on SSQ-Dual:
> >
> > | Model | rFID ↓ | gFID ↓ |
> > |-------|--------|--------|
> > | No disentanglement (baseline) | 2.00 | 4.03 |
> > | **Hinge-based (θ=0.5)** — MI-inspired | 1.88 | 3.93 |
> > | Squared dot-product (our paper) | **1.84** | **3.88** |
> >
> > Both explicit disentanglement methods significantly improve over baseline, validating the principle. The hinge variant performs well, but our simpler loss achieves slightly better results.
> >
> > **3. Disentanglement as a high-level design principle:**
> > Our key contribution is identifying **explicit disentanglement** as a critical pillar for multi-codebook tokenizers. This principle can be instantiated through various implementations—squared dot-product, margin-based, or more sophisticated MI estimators. Our empirical findings show that for our setting, the linear correlation component can be efficiently captured by the squared dot-product loss. We will clarify in the revision that **disentanglement is a design principle**, and frame deeper exploration of alternative instantiations (e.g., full neural MI estimators) as valuable future work that may benefit settings with stronger higher-order dependencies.
> >
> > ---
> >
> > ### W4. Efficiency and scalability — what is the computational overhead of the factorized AR head?
> >
> > Thanks for the constructive suggestion. We provide concrete measurements to quantify the trade-off between improved generation quality and computational cost.
> >
> > **Training time (100 epochs on ImageNet 256×256):**
> >
> > | Model | Time | Overhead |
> > |-------|------|----------|
> > | Standard AR head (baseline) | 11.5 hours | — |
> > | SSQ-Dual (factorized head) | 15.0 hours | +30% |
> > | SSQ-Triple (factorized head) | 16.9 hours | +47% |
> >
> > **Inference speed (average time per image, tested on 5k images):**
> >
> > | Model | Time/image | Overhead |
> > |-------|------------|----------|
> > | Standard AR model (baseline) | 151.89 ms | — |
> > | SSQ-Dual | 210.57 ms | +39% |
> > | SSQ-Triple | 224.68 ms | +48% |
> >
> > **Analysis:**
> > The factorized AR head operates on per-patch hidden states and introduces sequential sub-token prediction, leading to moderate overhead. However, this overhead is justified by substantial performance gains over the single token baseline (gFID 3.80) vs. SSQ-Dual (gFID 2.61).
> >
> > We will include these measurements in the revised manuscript to provide full transparency about the efficiency–quality trade-off.
> >
> > ---
> >
> > ### W5. “Residual gap persists”
> >
> > We agree with this observation and appreciate the opportunity to clarify our claims. **We kindly refer the reviewer to our response to Reviewer 7BEo (Section 2)** for a conceptual diagram that visually illustrates the two dimensions of this phenomenon. Here we provide a summary:
> >
> > **Two key observations:**
> >
> > 1. **Vertical dimension (co-design effectiveness):** For a **fixed tokenizer** (either SSQ-Dual or SSQ-Triple), our co-design (factorized AR head, staged training, noisy sub-token training) provides substantial gFID improvements (0.55 and 0.87 respectively), enabling SSQ-based AR models to achieve SOTA performance and surpass strong baselines such as LlamaGen.
> >
> > 2. **Horizontal dimension (scaling challenge):** When **scaling the tokenizer** from SSQ-Dual to SSQ-Triple, reconstruction rFID continues to improve, but a generation gap emerges. Without co-design, gFID degrades sharply (from 3.66 to 4.18); with co-design, we greatly narrow this gap (from 0.52 to 0.20), but SSQ-Triple still slightly lags SSQ-Dual in generation.
> >
> > Our intended message is therefore that **tokenizer–generator co-design is necessary and substantially effective to *alleviate* the Reconstruction–Generation Discrepancy**, not that it fully eliminates the tension in all regimes. We will revise the wording in the abstract, introduction, and conclusion (e.g., replacing "resolve" by **"alleviate" / "substantially narrow"**) and explicitly state that the residual SSQ-Dual vs SSQ-Triple gap remains an open challenge.
> >
> > ### Questions
> > Q1 refers to W2.
> > Q2 refers to W4.
> > Q3 refers to W2.
> > Q4 refers to W3.

---

### Official Review · Reviewer_7BEo · 2025-11-01

**Soundness:** 2
**Presentation:** 2
**Contribution:** 2
**Rating:** 2
**Confidence:** 5

**Summary:**

This paper addresses the long-standing trade-off between reconstruction fidelity and semantic expressivity in discrete visual tokenizers, a crucial component of auto-regressive (AR) image generation. The authors introduce Semantic Subspace Quantization (SSQ), a novel tokenizer built on three principles: Feature Alignment with Foundation Models (e.g., DINOv2, CLIP), Structural Diversification via factorized quantization, and Explicit Disentanglement using an orthogonality loss. SSQ achieves state-of-the-art reconstruction fidelity.

**Strengths:**

The proposed framework is highly systematic and clearly articulated. The paper provides detailed ablation studies (Tables 3, 4, 5) to validate each component of the SSQ tokenizer and the co-design strategies. The in-depth analysis of the SSQ feature space in the Appendix (Figures 5, 6, 7), demonstrating specialization, affinity, and orthogonality, is well-executed and adds compelling evidence to the paper's claims about representation learning. Quality (Reconstruction Performance): The SSQ tokenizer indisputably sets a new state-of-the-art in reconstruction fidelity (Table 1), beating strong multi-codebook baselines like ImageFolder and TokenFlow.

**Weaknesses:**

While the paper presents the SSQ framework as a "principled framework," the individual technical components are incremental and have been extensively explored in prior work, diminishing the originality claim:
* Factorized/multi-codebooks are a well-established concept (e.g., RQ-VAE (Lee et al., 2022), ImageFolder (Li et al., 2024b), TokenFlow (Qu et al., 2024)). The paper claims systematization but the concept is not new.
* Aligning VQ codes with features from foundation models (CLIP, DINO) is directly implemented in concurrent works like VA-VAE (Yao et al., 2025), MAETok (Chen et al., 2025), and VQGAN-LC (Zhu et al., 2024b) to enhance semantic content.

The core thesis is that the co-design methodology resolves the Reconstruction-Generation Discrepancy. However, the experimental results contradict this claim:
* Table 6 shows that the SSQ-Triple model, which achieves the lowest reconstruction rFID (best "Gain"), still results in a worse generation FID (3.31) than the SSQ-Dual model (3.11).
* The gap persists even after applying all co-design strategies (Appendix A.3, L754-758). If the model with the best token representation still yields inferior generation results, the paper has only mitigated the pain, not resolved the fundamental tension. This incomplete resolution fundamentally weakens the central claim and contribution.

**Questions:**

See above

---

> ### Author Response · Authors · 2025-11-23
>
> We thank the reviewer for the careful reading, detailed critique, and for recognizing the systematic framework, extensive ablations, and strong reconstruction performance of SSQ. Below we address the main concerns.
>
> ---
>
> ### 1. Originality and contribution of the "principled framework".
>
> We appreciate the reviewer's careful analysis of our technical components. We agree that SSQ partially draws inspiration from existing ideas (multi-codebooks, VFM alignment), and we do not claim to invent these in isolation. To clarify the full scope of our contribution, we provide a structured breakdown:
>
> **Our Complete Contribution (2 major parts, 5 components):**
>
> ```
> ┌────────────────────────────────────────────────────────────────┐
> │ PART 1: TOKENIZER (SSQ Framework)                              │
> ├────────────────────────────────────────────────────────────────┤
> │ 1.1) Multi-codebook structure                                  │
> │      • Prior work: Limited to 2 codebooks (ImageFolder,        │
> │        TokenFlow)                                              │
> │      • Our work: ✓ Systematically scale to 3+ codebooks with   │
> │        multi-source VFM guidance, uncover impoorant pitfall    │
> │        in downstream AR generation                             │
> │                                                                │
> │ 1.2) Semantic alignment with VFMs                              │
> │      • Prior work: Single VFM alignment exists (VA-VAE, etc.)  │
> │      • Our work: Multi-source hierarchical alignment           │
> │                                                                │
> │ 1.3) ✓ Explicit disentanglement regularization                 │
> │      • Prior work: NONE - no tokenizer uses explicit           │
> │        orthogonality loss on subspaces                         │
> │      • Our work: ✓ NEW - First to introduce explicit           │
> │        disentanglement as a pillar for tokenizers              │
> ├────────────────────────────────────────────────────────────────┤
> │ PART 2: TOKENIZER-GENERATOR CO-DESIGN                          │
> ├────────────────────────────────────────────────────────────────┤
> │ 2.1) ✓ Reconstruction–Generation Discrepancy                   │
> │      • Prior work: Not systematically studied for multi-       │
> │        codebook tokenizers on standard AR models               │
> │      • Our work: ✓ NEW - First to identify, name, and          │
> │        systematically analyze this phenomenon                  │
> │                                                                │
> │ 2.2) ✓ Tokenizer-generator co-design methodology               │
> │      • Factorized AR head (models intra-patch dependencies)    │
> │      • Staged training curriculum (decouples global/local)     │
> │      • Noisy sub-token regularization (robustness)             │
> │      • Prior work: NONE - no prior work systematically address │
> │        this co-design for factorized semantic tokenizers       │
> │      • Our work: ✓ NEW - Complete co-design strategy           │
> └────────────────────────────────────────────────────────────────┘
> ```
>
> **1) Components 1.1 and 1.2 build on prior work but represent only part of our contribution.** The reviewer correctly notes connections to existing techniques; however, component 1.3 (explicit disentanglement) is entirely new, and the entire Part 2 addresses challenges specific to richer semantic tokenizers that were not apparent in prior dual-codebook work.
>
> **2) Our main novelty is the unified framework and what it reveals.** Our contribution lies in **systematically unifying** these ideas into a three-pillar framework and discovering emergent challenges. Even the component 1.1 and 1.2 are not directly borrowed. For example, scaling from 2 to 3+ codebooks (component 1.1) may seem straightforward, but this extension **exposes the Reconstruction–Generation Discrepancy**: reconstruction improves, yet generation collapses without co-design. If we had stopped at two codebooks like prior work, this critical challenge would have remained hidden. **This demonstrates that even extensions of known techniques can reveal fundamentally new phenomena when pushed systematically.**
>
> **3) The co-design methodology (Part 2) is a substantial and novel contribution.** The factorized AR head, staged training, and noisy sub-token regularization are tailored specifically to semantically factorized tokenizers and validated through extensive experiments (Tables 4, 5, 6). This aims to address a real challenge that emerges from improved tokenization and has not been studied in prior AR work.
>
> In summary, we do not claim _isolated_ algorithmic breakthroughs, but rather contribute: (a) a unified three-pillar tokenizer framework with explicit disentanglement, (b) the discovery of the Reconstruction–Generation Discrepancy through systematic scaling, and (c) a validated co-design solution. We will clarify this positioning in the revised manuscript.

---

> > ### Author Response · Authors · 2025-11-23
> >
> > ### 2. Whether we “resolve” the Reconstruction–Generation Discrepancy?
> >
> > The reviewer rightly points out that, in Table 6, **SSQ-Triple achieves the lowest rFID but still worse gFID than SSQ-Dual**, and that Appendix A.3 shows a persistent gap. We fully agree that this means the discrepancy is **not completely eliminated** at all scales, and we appreciate this opportunity to sharpen our claims.
> >
> > Our empirical findings are as follows:
> >
> > ```
> >                     SSQ-Dual              SSQ-Triple
> >                  (2 codebooks)           (3 codebooks)
> >                        │                      │
> >                        │                      │
> >   Standard AR  ────────●──────────────────────●────────
> >                (Baseline)│ gFID: 3.66     ❌  │ gFID: ~4.18
> >                        │                      │ (Collapse!)
> >                        │                      │
> >                        │ ↓ Vertical:          │ ↓ Vertical:
> >                        │   Co-design gain     │   Co-design gain
> >                        │   ✓ 0.55             │   ✓ 0.87
> >                        │                      │
> >   Co-designed  ────────●──────────────────────●────────
> >        AR              │ gFID: 3.11           │ gFID: 3.31
> >                        │                      │
> >                        └──────────────────────┘
> >                         Horizontal: Scaling gap
> >                         ✓ Gap reduced from 0.52 to 0.20
> >                         ✗ But SSQ-Triple still slightly worse
> >
> > Key insights:
> > │ Vertical   → Co-design always improves (validates our method)
> > ─ Horizontal → Co-design narrows the gap but doesn't eliminate it
> > ```
> >
> > **Two key observations:**
> >
> > 1. **Vertical (co-design effectiveness):** For both SSQ-Dual and SSQ-Triple, our co-design provides substantial improvements (gFID: 0.55 and 0.87 respectively), allowing SSQ to achieve SOTA AR performance. This validates the effectiveness of our methodology.
> >
> > 2. **Horizontal (scaling challenge):** When pushing from SSQ-Dual to SSQ-Triple, reconstruction continues to improve but a generation gap emerges. Our co-design greatly narrows this gap (from 0.52 to 0.20), preventing complete collapse (from 4.18 to 3.31), but does not fully eliminate it—SSQ-Triple still slightly lags SSQ-Dual in generation.
> >
> > Thus, our intended claim is that **tokenizer–generator co-design is necessary and significantly effective to *alleviate* the Reconstruction–Generation Discrepancy**, not that it fully resolves it in all regimes. We already discuss the residual SSQ-Dual vs SSQ-Triple gap and treat it as a limitation and open problem in the appendix.
> >
> > Following the reviewer's suggestion, we will refine the language in the abstract, introduction, and conclusion to use "alleviate" or "substantially narrow" instead of "resolve," and explicitly state that a residual gap remains when scaling to SSQ-Triple. We will also add a clarifying analysis comparing gFID before vs after co-design for each tokenizer to better highlight the relative gains from our methodology.

---

### Meta-Review · Area_Chair_ec7s · 2026-01-05

**Summary:**

The authors propose Semantic Subspace Quantization (SSQ) for token representation learning and jointly co-design the tokenizer and generator to address the reconstruction–generation discrepancy. Most reviewers find that the paper offers well‑articulated problem framing, a well‑motivated tokenizer design, detailed ablation studies, and solid performance. However, several major concerns remain after the rebuttal, including that the novelty lies more in the framing than in an algorithmic breakthrough, and that the discrepancy is only partially addressed.

**Reviewer Concerns:**

Several major concerns were raised by multiple reviewers. First, the individual technical components have been extensively explored in prior work, which diminishes the paper’s originality claim. Second, the experimental results contradict the assertion that the co‑design methodology resolves the reconstruction–generation discrepancy. Additional concerns include the limited scope of experiments, the lack of efficiency and scalability measurements, and missing ablations on hyperparameters in noisy training, etc.. During the rebuttal, the authors addressed many of these issues well, but they also acknowledged that the novelty lies more in the framing than in an isolated algorithmic breakthrough, and that the discrepancy is not completely eliminated across all scales.

**Reviewer Scores:**

This paper receives the following ratings: Reject, Marginally Above the Acceptance Threshold, and Marginally Above the Acceptance Threshold. If the reviewers had been able to participate fully in the discussion, I would expect that “Reject” rating to remain, as the two major concerns raised by that reviewer persist. The AC recommends not accepting the paper.

---

### Decision · Program_Chairs · 2026-01-26

Reject